# A Semi-Quantum Private Comparison Base on W-States

**DOI:** 10.3390/e25091269

**Published:** 2023-08-29

**Authors:** Jian Li, Zhuo Wang, Jun Yang, Chongqiang Ye, Fanting Che

**Affiliations:** 1School of Information Engineering, Ningxia University, Yinchuan 750021, China; lijian@bupt.edu.cn (J.L.); dragon@nxu.edu.cn (J.Y.); 2School of Cyberspace Security, Beijing University of Posts and Telecommunications, Beijing 100876, China; 3School of Artificial Intelligence, Beijing University of Posts and Telecommunications, Beijing 100876, China; chongqiangye@bupt.edu.cn (C.Y.); qingxi1999@outlook.com (F.C.)

**Keywords:** quantum private comparison, quantum cryptography, quantum communication

## Abstract

Privacy comparison is an important research topic in secure multi-party computing, widely used in e-commerce, secret ballots, and other fields. However, the development of quantum computing power poses a growing potential security threat to secure multi-party algorithms based on mathematically tricky problems, and most of the proposed quantum privacy comparison schemes could be more efficient. Therefore, based on the W-state, we offer a more efficient semi-quantum privacy comparison method. The security analysis shows that the scheme can resist third-party, measurement, and entanglement attacks. Compared with the previous work, the scheme significantly improves communication efficiency and has stronger practicability.

## 1. Introduction

Secure multiparty computing (secure multiparty computation, SMC) was proposed by Yao to prevent multiple participants’ privacy leakage and let them work together to solve the computing problem [1]. SMC is widely used in e-commerce, data compression, and secret ballots. Secure multiparty computing (SMC) is an essential topic in distributed computing. It allows a group of users who do not trust each other to perform distributed computing with the participation of a semi-trusted third party to obtain the comparison results of private information without revealing their input.

However, the quantum computer threatens the security of classical SMC. With the improvement of computing power and the emergence of quantum algorithms [2,3], the SMC cryptographic protocol based on classical NP problems is repeatedly broken. As an indivisible smallest unit, quantum has its unique properties. The principle of quantum uncertainty makes it impossible for eavesdroppers to measure the state of transmitted quantum states accurately, and the principle of quantum non-cloning ensures that eavesdroppers cannot accurately copy transmitted quantum states and obtain adequate information [4,5,6]. Therefore, quantum privacy comparison is applied to resist quantum attacks as a branch of quantum cryptography.

Compared with quantum communication, semi-quantum communication has unique advantages, which are easier to realize while ensuring security. Users can use only straightforward quantum devices, saving the high cost of buying or preparing quantum states. In particular, if the device fails during quantum communication, it can switch from quantum communication to semi-quantum communication to complete the whole process. Therefore, the research on semi-quantum cryptographic communication is significant in quantum communication. Compared with quantum privacy comparison, semi-quantum privacy comparison has the advantages of lower requirements on hardware devices, easier realization, and more common application scenarios.

In 2016, the first semi-quantum privacy comparison protocol (SQPC) [7] based on Bell states was proposed by Chou et al., ensuring that two classical users compare their private information without revealing privacy. Subsequently, many improvements and optimizations of SQPC have been proposed [8,9,10,11,12,13,14,15,16,17,18]. In 2018, Ye et al. [11] constructed an SQPC protocol based on the product of two-particle tensors without an entanglement exchange. In 2018, Thapliyal et al. [12] proposed an SQPC protocol based on an orthogonal state in a noisy environment, which adopted the Bell state as a quantum state carrier In the same year, Lang [13] proposed two SQPC protocols with different TP identities and abandoned the quantum entanglement exchange operation. For malicious TP, he cannot know the private information or the results. Lin et al. [14] abandoned the quantum entanglement exchange operation in 2019. They used a single-photon state to build a practical and efficient SQPC protocol, requiring a shared key in advance to achieve. In 2021, Tian et al. [15] proposed a novel semi-quantum private comparison (SQPC) protocol based on W-state, which is more efficient. In 2022, Wang et al. [16] designed an SQPC protocol using the GHZ state with D-dimension. Geng et al. [17] constructed the SQPC protocol for the D-level single-particle state size relationship. In 2023, He et al. [18] proposed an improved SQPC protocol with a higher security level than Tian et al. ‘s scheme [15]. Based on the decoherence-free states, two multi-party semi-quantum private comparison protocols are proposed to counteract collective noises [19]. And by adopting d-dimensional Bell states, Lian et al. [20] constructed an MQPC protocol that can be used in a strange user environment.

In this work, a semi-quantum privacy comparison (SQPC) protocol is proposed, which has higher efficiency compared with before works; it does not need to share the key in advance and has a higher interception detection rate. Therefore, this scheme is more practical. The structure of this paper is as follows: in Section 2, the work related to semi-quantum privacy comparison and the basis of quantum information are introduced; in Section 3, the steps and contents of the protocol are explained in detail; the security analysis part is shown in the fourth part; finally, the summary of this work is given.

## 2. Preliminaries

### 2.1. Entangled States

The two W-states were used to distribute to the three participants, and they are denoted by
(1)$$|W\rangle =\frac{1}{\sqrt{3}}\left(|011\rangle +|010\rangle +|100\rangle \right),$$
(2)$$|H\rangle =\frac{1}{2}\left(|001\rangle +|010\rangle +|100\rangle +|111\rangle \right),$$

The |W〉 state can be described as
(3)$$\begin{array}{l} |W\rangle =\sqrt{\frac{2}{3}}\left({\frac{|01\rangle +|10\rangle }{\sqrt{2}}}_{1,2}⊗{|0\rangle }_{3}\right)+\frac{1}{\sqrt{6}}{\left(\frac{|00\rangle +|11\rangle }{\sqrt{2}}+\frac{|00\rangle -|11\rangle }{\sqrt{2}}\right)}_{1,2}⊗{|1\rangle }_{3}\\ =\sqrt{\frac{2}{3}}\left({\frac{|01\rangle +|10\rangle }{\sqrt{2}}}_{1,3}⊗{|0\rangle }_{2}\right)+\frac{1}{\sqrt{6}}{\left(\frac{|00\rangle +|11\rangle }{\sqrt{2}}+\frac{|00\rangle -|11\rangle }{\sqrt{2}}\right)}_{1,3}⊗{|1\rangle }_{2}\\ =\sqrt{\frac{2}{3}}\left({\frac{|01\rangle +|10\rangle }{\sqrt{2}}}_{2,3}⊗{|0\rangle }_{1}\right)+\frac{1}{\sqrt{6}}{\left(\frac{|00\rangle +|11\rangle }{\sqrt{2}}+\frac{|00\rangle -|11\rangle }{\sqrt{2}}\right)}_{2,3}⊗{|1\rangle }_{1}\\ =\sqrt{\frac{2}{3}}\left(|\psi^{+}\rangle \right)⊗|0\rangle +\frac{1}{\sqrt{6}}\left(|\phi^{+}\rangle +|\phi^{-}\rangle \right)⊗|1\rangle \end{array},$$

Therein, the subscript marks the particle position.

The |H〉 state can be described as
(4)$$\begin{array}{l} |H\rangle =\frac{1}{\sqrt{2}}\left(\frac{|00\rangle +|11\rangle }{\sqrt{2}}⊗|1\rangle +\frac{|01\rangle +|10\rangle }{\sqrt{2}}⊗|0\rangle \right)\\ =\frac{1}{\sqrt{2}}\left(|\phi^{+}\rangle ⊗|1\rangle +|\psi^{+}\rangle ⊗|0\rangle \right) \end{array},$$

Accordingly, the following circuit is used to implement |H〉 in Figure 1.

Among them, $|\psi^{+}\rangle$, $|\phi^{+}\rangle$, $|\phi^{-}\rangle$ are kinds of Bell states, which will be used to take the joint measurement and can be expressed a
(5)$$|\phi^{+}\rangle =\frac{1}{\sqrt{2}}\left(|00\rangle +|11\rangle \right),\;|\phi^{-}\rangle =\frac{1}{\sqrt{2}}\left(|00\rangle -|11\rangle \right),\;|\psi^{+}\rangle =\frac{1}{\sqrt{2}}\left(|01\rangle +|10\rangle \right).$$

### 2.2. One-Time Pad

The one-time pad (OTP) is a symmetric encryption algorithm, which was first invented by Frank Miller [21]. In OTP, the message sender and the message receiver share a random secret key k, called a one-time pad. In general, it requires that the random pad k and the message m to be sent should have the same size. To encrypt the message, the sender calculates c=m⊕k and sends OTP ciphertext c to the receiver, where the symbol “⊕” denotes addition under modular two. To decrypt c, the receiver calculates m=c⊕k. It is impossible for an adversary to break m from the ciphertext c due to the unconditional security of OTP [22,23].

## 3. Semi-Quantum Private Comparison Scheme

By default, the semi-quantum privacy comparison (SQPC) means two participants who do not have complete quantum capabilities are helped by a trusted third party (TP). As a semi-trusted third party, TP will not collude with participants to obtain users’ privacy, nor will it disclose any participants’ privacy; they have the full quantum capability to prepare, manipulate and measure quantum states. A new semi-quantum privacy comparison protocol based on three-particle entangled states is proposed below, which does not require a pre-shared key and has higher efficiency. The model of SQPC is shown in Figure 2.

Suppose that two copies of private information from Alice and Bob need to be compared. TP must compare whether its private information is consistent without knowing the specific information. Alice and Bob establish a quantum channel and classical channel with TP, respectively, and can only do the following two operations for the received qubit.

(1)MEASURE: Using the Z-basis to measure the qubit, re-prepare the same one with the measurement result and send it back.(2)REFLECT: Return the qubit without doing anything.

**Step 1:** TP prepares *N* (= 4 n) entangled states according to Equations (1) and (2), including 2 *n*
|W〉 and 2 *n*
|H〉. TP then divides these entangled states into three single particles. Separately, the particle at the first position forms the sequence $S_{A}=\left(s_{A}^{1},s_{A}^{1},\ldots s_{A}^{n}\right)$, the particle at the second position forms the sequence $S_{B}=\left(s_{B}^{1},s_{B}^{2},\ldots s_{B}^{n}\right)$, and the particle at the third position forms the sequence $S_{T}=\left(s_{T}^{1},s_{T}^{2},\ldots s_{T}^{n}\right)$; $S_{A}$, $S_{B}$, and $S_{T}$ are random quantum sequences of |0〉 or |1〉 from the |W〉 and |H〉 states; $S_{A}^{i}$, $S_{B}^{i}$ and $S_{T}^{i}$ represents the first state, the second state, and the third state, respectively.**Step 2:** TP sends the particle in the travel sequence $S_{A}$ to Alice and the state in the travel sequence $S_{B}$ to Bob, while it remains in TP’s own hands. Note that only TP knows whether the states come from |W〉 or from |H〉.**Step 3:** Alice and Bob randomly perform both MEASURE and REFLECT operations when receiving $S_{A}$ and $S_{B}$. Note that the probability that Alice and Bob choose both the MEASURE operation and the REFLECT operation is 1/2. Therein, Alice and Bob each select *n* quantum states to perform the MEASURE operation and select *n* quantum states to perform the REFLECT operation. The sequence sent back by Alice and bob is noted as ${\hat{S}}_{A}$ and ${\hat{S}}_{B}$. Meanwhile, according to the measurement results of MEASURE operation, Alice and Bob record 1 if it is |1〉; otherwise, if the outcome is |0〉, they record it as 0.**Step 4:** TP yields a new sequence of states after receiving all the sequences returned from Alice and Bob, respectively. Consequently, Alice and Bob each announce the position that is performed by the MEASURE operation. TP performs channel detection based on Alice’s and Bob’s operations. Specific operations are as follows: 

TP selects the pairs in which Alice and Bob both perform the REFLECT operation, uses the Bell basis joint measurement ($S_{A}^{i}$, $S_{B}^{i}$), and uses Z-basis to measure the retained in his own hand. Then, TP checks to see if the joint measurements match the state in $S_{T}^{i}$, that is, when the $S_{T}^{i}$ is from |W〉, the joint measurement result should be either $|\phi^{+}\rangle$ or $|\phi^{-}\rangle$; conversely, when the $S_{T}^{i}$ is from |H〉, the joint measurement result must only be $|\phi^{+}\rangle$. In particular, if the measurement in |H〉 appears $|\phi^{-}\rangle$, then the channel has an eavesdropper.

**Step 5:** After the channel detection is complete, TP picks out the particles that both Alice and Bob choose the MEASURE operation. In this case, Alice’s and Bob’s measurements may agree or not, but TP cannot determine it by measurement outcome. TP makes a Z-basis measurement for $S_{T}^{i}$ at these locations and publishes the measurement outcome, the published result in the corresponding position is marked 1 or 0. If the measurement is |0〉, it is recorded as 0. Otherwise, it is recorded as 1.**Step 6:** Alice and Bob negotiate whether TP is honest according to the position published by TP; for the position marked 1 by TP, Alice and Bob check that the measurements are consistent, and conversely, for the position marked 0, Alice and Bob check that the measurements are opposite. If the results show that the TP is honest, Alice and Bob share the key $K_{AB}$ through negotiation. When the measurement results of both parties are the same, $K_{AB}$ is the default value, that is the measurement result; when the measurement results are different, $K_{AB}$ is the random value, that is 0 or 1.**Step 7:** Alice and Bob share the key with TP, respectively. For Alice and Bob take different operations, Alice and Bob keep a secret sequence of their measurements denoted as $K_{AT}$, $K_{BT}$. Concretely, when Alice adopts MEASURE and Bob adopts REFLECT, TP adopts the Bell base joint measurement for ($S_{A}^{i}$, $S_{T}^{i}$). When Alice adopts REFLECT and Bob adopts MEASURE, TP adopts the Bell base joint measurement for ($S_{B}^{i}$, $S_{T}^{i}$). According to Equations (2) and (3), if the measurement is $|\phi^{+}\rangle$ or $|\phi^{-}\rangle$, TP yields $K_{AT}$ or $K_{BT}$ 1; otherwise, if the measurement result is $|\psi^{+}\rangle$, TP yields $K_{AT}$ or $K_{BT}$ 0.**Step 8:** TP implements privacy protection as follows. Suppose the private messages to be compared from Alice and Bob are $m_{A}=\left[m_{a1},m_{a2},\ldots m_{an}\right]$, $m_{B}=\left[m_{b1},m_{b2},\ldots m_{bn}\right]$. Alice computes
(6)$$C_{A}=m_{A}⊕K_{AT}⊕K_{AB},$$

Bob computes
(7)$$C_{B}=m_{B}⊕K_{BT}⊕K_{AB},$$

Alice and Bob send the computation result to TP, respectively.

Immediately after, TP computes
(8)$$C=C_{A}⊕C_{B}⊕K_{AT}⊕K_{BT}.$$

According to the law of modular two operations, if C is a bit sequence of 0, it means the privacy information is the same; if 1 appears in the sequence, it means the privacy information is different. Below, the operations and purposes of different situations in the protocol are summarized in Table 1.

## 4. Security and Efficiency Analysis

This chapter first gives the correctness analysis, then analyzes the security of semi-trusted third parties and users, and finally provides the attack with the analysis of malicious users.

### 4.1. Correctness

According to Equations (6) and (7), Equation (8) is can be written as
(9)$$\begin{array}{l} C=(m_{A}⊕K_{AT}⊕K_{AB})⊕(m_{B}⊕K_{BT}⊕K_{AB})⊕K_{AT}⊕K_{BT}\\ =m_{A}⊕m_{B} \end{array},$$

Therefore, they will satisfy the following equality,
(10)$$\{\begin{array}{l} C=m_{A}⊕m_{B}=1,m_{A}\neq m_{B}\\ C=m_{A}⊕m_{B}=0,m_{A}=m_{B} \end{array}.$$

According to Equation (10), every bit of the binary sequence is performed modulo 2; if it is the same binary number in that position, the result is 0; otherwise it is 1.

To sum up, the privacy comparison of the protocol can be prepared to determine whether the binary sequence is consistent through the calculation results. If 1 appears in succession, it means that the information in this position is inverted; if C is a string of 0, it means that the two pieces of information are the same.

### 4.2. Outside Attack

Generally, there are three attack ways for external attackers to steal users’ privacy information: intercept re-transmission attack, measurement-re-transmission attack, and entanglement measurement attack. For each type of attack, the following section provides a corresponding security analysis.

#### 4.2.1. Intercept Re-Transmission Attack

Take the quantum channel of Alice and TP as an example. Assuming that Eve, as a fake TP, intercepts the sequence sent by TP to Alice, Eve sends Alice the fake sequence prepared in advance. Alice randomly performs MEASURE operations and REFLECT operations on the fake sequence. According to the protocol design, she will send the sequence to Eve (fake TP). Later, Alice and Bob will announce where they each took the non-stop action. At this point, Eve can use Alice’s and Bob’s disclosure to create a false key, denoted ${K^{\prime }}_{AT}$ and ${K^{\prime }}_{BT}$ respectively. However, Eve still does not have access to Alice and Bob’s private information. According to protocol Step 8, Equations (6) and (7), even if Eve obtains Alice’s and Bob’s calculation results as a fake TP, he cannot obtain the privacy information $m_{A}$ and $m_{B}$ from the calculation results according to the one-time-secret nature of OTP [21,22,23]. To make matters worse, TP will discover Eve’s presence by comparing fake sequences forwarded by Eve. Furthermore, since the qubits corresponding to the false sequence and the issuing sequence are different, TP and Alice or Bob cannot complete the shared key stage. According to Equations (6) and (7), TP cannot decrypt $C_{A}$ and $C_{B}$ through $K_{AT}$ and $K_{BT}$. At this time, TP will again determine that there is an attacker intercepting the particles of the channel.

#### 4.2.2. Measure Re-Transmission Attack

In the measurement re-transmission attack, the attacker intercepts the TP sequence sent to the user, measures it, and forwards it. Taking Alice’s channel as an example, it is assumed that Eve intercepts the sequence $S_{A}$ and makes a Z-basis measurement of its particles, and then Eve sends the measured sequence $S_{A}$ to Alice. According to the physical properties of the collapse measured by the entangled particles, Alice’s random operation will not change the state of the particles. Therefore, when Alice sends the particle back to TP, TP will find Eve’s attack behavior through the detection of Step 4.

Compared with the previous scheme, our scheme will have a higher interception detection rate. If Alice chooses the MEASURE operation and Bob chooses the REFLECT operation, the attack will not be detected. If Alice performs the MEASURE operation, Bob performs the REFLECT operation, and TP performs the Bell measurement and single measurement; then, if the single particle measurement is |0〉, the attack behavior is detected with a probability of 1/4, if the single particle measurement is |1〉, the probability in |W〉 and |H〉 are 1/2 and 1/4. If Alice and Bob both take the return operation, TP compares the relationship between the Bell measurement and the single particle measurement; then, if the single particle measurement is |0〉, the attack behavior is detected with a probability of 1/4, and if it is |1〉, the probability in |W〉 and |H〉 is 1/2 and 1/4. Therefore, the probability that Eve’s attack behavior is detected under the measurement of repeated attacks is
(11)$$\begin{array}{l} p=\frac{1}{2}\left[\frac{1}{3}\left(\frac{1}{4}\times \frac{1}{4}+\frac{1}{4}\times \frac{1}{4}\right)+\frac{2}{3}\left(\frac{1}{4}\times \frac{1}{2}+\frac{1}{4}\times \frac{1}{2}\right)\right]\\ +\frac{1}{2}\left[\frac{1}{2}\left(\frac{1}{4}\times \frac{1}{4}+\frac{1}{4}\times \frac{1}{4}\right)+\frac{1}{2}\left(\frac{1}{4}\times \frac{1}{2}+\frac{1}{4}\times \frac{1}{2}\right)\right]\\ =\frac{19}{96} \end{array},$$

In this regard, the eavesdropping detection probability of n particle length sequence is $1-{\left(\frac{19}{96}\right)}^{n}$, as n increases, it approaches 1.

#### 4.2.3. Entanglement Measurement Attack

It is possible for an attacker to construct a new quantum system by introducing auxiliary particles to avoid eavesdropping detection and monitor the whole system by measuring other particles. To achieve the purpose of obtaining private information.

A group of auxiliary particles is introduced to construct a two-dimensional Hilbert space, and a group of orthonormal basis is selected to describe the space vector, which can be expressed as $|\tau_{00}\rangle$, $|\tau_{01}\rangle$, $|\tau_{10}\rangle$, $|\tau_{11}\rangle$. To distinguish the source of the particles, we will represent Alice’s particles from |W〉 and |H〉 as {${|0\rangle }_{AW}$, ${|1\rangle }_{AW}$, ${|0\rangle }_{AH}$, ${|1\rangle }_{AH}$}, and Bob’s particles from |W〉 and |H〉 as {${|0\rangle }_{BW}$, ${|1\rangle }_{BW}$, ${|0\rangle }_{BH}$, ${|1\rangle }_{BH}$}.

For a particle in the |H〉, ${|0\rangle }_{H}$ represents auxiliary particles. The unitary operations performed on qubits.
(12)$$U_{H}({|0\rangle }_{AH}{|0\rangle }_{H})={|0\rangle }_{AH}|\tau_{00}\rangle +{|1\rangle }_{AH}|\tau_{01}\rangle ,$$
(13)$$U_{H}({|1\rangle }_{AH}{|0\rangle }_{H})={|0\rangle }_{AH}|\tau_{10}\rangle +{|1\rangle }_{AH}|\tau_{11}\rangle .$$

Then, the whole system space can be expressed as
(14)$$\begin{array}{l} ({|0\rangle }_{AH}|\tau_{00}\rangle +{|1\rangle }_{AH}|\tau_{01}\rangle ){|0\rangle }_{BH}+({|0\rangle }_{AH}|\tau_{10}\rangle +{|1\rangle }_{AH}|\tau_{11}\rangle ){|1\rangle }_{BH}\\ ={|00\rangle }_{ABH}|\tau_{00}\rangle +{|10\rangle }_{ABH}|\tau_{01}\rangle +{|01\rangle }_{ABH}|\tau_{10}\rangle +{|11\rangle }_{ABH}|\tau_{11}\rangle \\ =|\phi^{+}\rangle \left(|\tau_{00}\rangle +|\tau_{11}\rangle \right)+|\phi^{-}\rangle \left(|\tau_{00}\rangle -|\tau_{11}\rangle \right)+\left(|\psi^{+}\rangle -|\psi^{-}\rangle \right)|\tau_{01}\rangle +\left(|\psi^{+}\rangle +|\psi^{-}\rangle \right)|\tau_{10}\rangle \end{array},$$

As can be seen from the above equation, Eve’s attacks must meet the following conditions in order to remain undetected
(15)$$|\tau_{01}\rangle +|\tau_{10}\rangle =0,$$
(16)$$|\tau_{00}\rangle -|\tau_{11}\rangle =0.$$

Therefore, for the particles from |W〉, Eve cannot obtain any information from the state of the auxiliary particle, because its state is independent of the state of the other particles. 

For a particle in the |W〉, $U_{W}$ represents unitary operations performed on qubits.
(17)$$U_{W}{|0\rangle }_{AW}={|0\rangle }_{AW}|\tau_{00}\rangle +{|1\rangle }_{AW}|\tau_{01}\rangle ,$$
(18)$$U_{W}{|1\rangle }_{AW}={|0\rangle }_{AW}|\tau_{10}\rangle +{|1\rangle }_{AW}|\tau_{11}\rangle ,$$

At this point, the whole system can be described as
(19)$$\begin{array}{l} |W\rangle =\frac{1}{\sqrt{3}}\left({|0\rangle }_{AW}|\tau_{00}\rangle +{|1\rangle }_{AW}|\tau_{01}\rangle \right){|01\rangle }_{BTW}+\left({|0\rangle }_{AW}|\tau_{00}\rangle +{|1\rangle }_{AW}|\tau_{01}\rangle \right){|10\rangle }_{BTW}\\ +\left({|0\rangle }_{AW}|\tau_{10}\rangle +{|1\rangle }_{AW}|\tau_{11}\rangle \right){|00\rangle }_{BTW}\\ =\frac{1}{\sqrt{3}}\left(\begin{array}{l} {|01\rangle }_{ABW}|\tau_{00}\rangle {|0\rangle }_{TW}+{|11\rangle }_{ABW}|\tau_{01}\rangle {|0\rangle }_{TW}+{|00\rangle }_{ABW}|\tau_{00}\rangle {|1\rangle }_{TW}\\ +{|00\rangle }_{ABW}|\tau_{10}\rangle {|0\rangle }_{TW}+{|10\rangle }_{ABW}|\tau_{01}\rangle {|1\rangle }_{TW}+{|10\rangle }_{ABW}|\tau_{11}\rangle {|0\rangle }_{TW} \end{array}\right)\\ =\frac{1}{\sqrt{6}}\left[\begin{array}{l} \left({\left(|\phi^{+}\rangle +|\phi^{-}\rangle \right)}_{ABW}|\tau_{00}\rangle +{\left(|\psi^{+}\rangle -|\psi^{-}\rangle \right)}_{ABW}|\tau_{01}\rangle \right){|1\rangle }_{TW}\\ +{\left(|\psi^{+}\rangle +|\psi^{-}\rangle \right)}_{ABW}|\tau_{00}\rangle +{\left(|\phi^{+}\rangle -|\phi^{-}\rangle \right)}_{ABW}|\tau_{01}\rangle +\\ {\left(|\psi^{+}\rangle +|\psi^{-}\rangle \right)}_{ABW}|\tau_{11}\rangle {|0\rangle }_{TW}+{\left(|\phi^{+}\rangle +|\phi^{-}\rangle \right)}_{ABW}|\tau_{10}\rangle \end{array}\right] \end{array},$$

Therefore, in order for Eve to pass eavesdropping detection, it needs to meet
(20)$$|\tau_{00}\rangle =|\tau_{01}\rangle =|\tau_{01}\rangle =|\tau_{11}\rangle =0.$$

However, Eve will not be able to distinguish between Alice’s measurements and will not be able to obtain useful information.

### 4.3. Inside Attack

Without loss of generality, Bob is a dishonest actor, assuming he is trying to obtain Alice’s privacy while making a private comparison with her. Bob may take any attack form of the external attacker to obtain information. Unlike the external attacker, Bob has the same key $K_{AB}$ with Alice, so as long as he can obtain the correct $K_{AT}$, he can really get the private information. However, in this case, he cannot obtain the correct and valid $K_{AT}$, If Bob takes a measurement re-transmission attack to obtain K, Bob has no way of knowing what specific operation Alice has chosen. Alice has a 1/2 probability of measuring the quantum state, so once Bob returns all the intercepted $S_{A}$, there is a 1/2 probability of choosing the wrong operation. At this time, the interception detection rate is 1/4, so the sequence’s interception detection rate is $1-{\left(\frac{3}{4}\right)}^{n}$, When *n* is large enough, the eavesdropping detection rate will approach 1.

On the other hand, if Bob tries to attack by measuring re-transmissions or entanglement measurements, he will be regarded as an external attacker. As analyzed in Section 4.2.2 and Section 4.2.3, any attack strategy will inevitably introduce a certain error rate and fail to obtain any information.

### 4.4. Efficiency Analysis

The SQPC protocol efficiency calculation method is proposed in [24], which can be described as
(21)$$\eta =\frac{c}{q+b},$$
where the c represents the classical particle of comparison, q represents the total qubits used for exchange, and b represents the qubit used by the two participants.

The SQPC protocol uses a three-particle pure state for privacy comparisons. TP prepared 12 qubits and sent them to two participants. When Alice and Bob measured the particles, they each prepared 1 qubit. Hence the scheme of efficiency is
(22)$$\eta =\frac{1}{12+2}\approx 0.07143$$

Here, as shown in Table 2, a comparison table of schemes is given to show the preponderance of this work in the following.

Compared with the previous scheme, the proposed scheme does not require pre-shared keys, and does not consume resources for key sharing; moreover, it improves qubit efficiency.

## 5. Conclusions

A secure and efficient SQPC protocol based on W-state is proposed. Two classic participants can compare the equality of their private information with the help of a semi-honest assistant named TP. Alice and Bob cannot know each other’s privacy information, and the semi-honest third-party TP cannot obtain any privacy information compared between Alice and Bob. The proposed protocol can resist attacks and guarantee high efficiency and a high interception detection rate. Moreover, it doubles the bit efficiency of previous protocols.

## Figures and Tables

**Figure 1 entropy-25-01269-f001:**
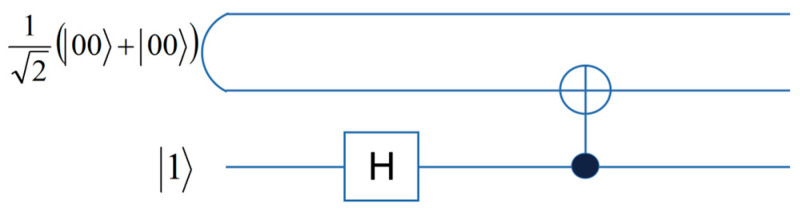
The preparation circuit of |H〉.

**Figure 2 entropy-25-01269-f002:**
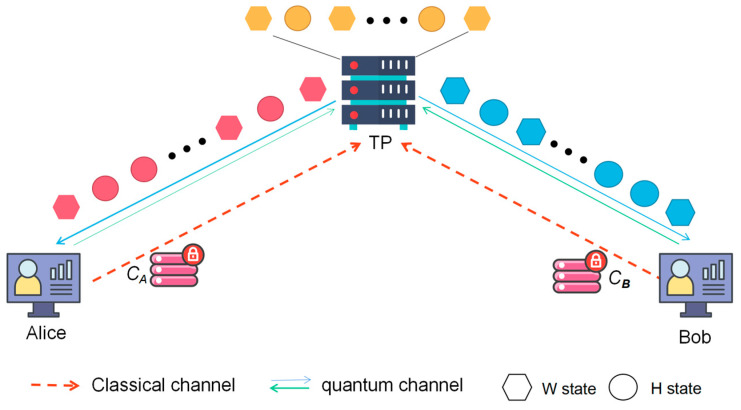
The model of protocol.

**Table 1 entropy-25-01269-t001:** Summary table of state types and their corresponding operations.

State	Operation (Alice)	Operation (Bob)	Purpose	Population
|W〉	REFLECT	REFLECT	Detection channel	*n*/2
|H〉	REFLECT	REFLECT		*n*/2
|W〉	MEASURE	REFLECT	Share Key	*n*/2
|H〉	MEASURE	REFLECT		*n*/2
|W〉	REFLECT	MEASURE		*n*/2
|H〉	REFLECT	MEASURE		*n*/2
|W〉	MEASURE	MEASURE	Comparison	*n*/2
|H〉	MEASURE	MEASURE		*n*/2

If T is a bit.

**Table 2 entropy-25-01269-t002:** The comparison of related work.

SQPC Protocol	Ref. [8]	Ref. [11]	Ref. [9]	Ref. [10]	Our Scheme
Quantum states	Bell	Single photon	GHZ	GHZ-like	W-state
Consumption of key sharing	0	16*n*	24*n*	0	0
Consumption of comparison	160*n*	24*n*	8*n*	32*n*	14*n*
Consumption of communication	160*n*	40*n*	32*n*	32*n*	14*n*
Qubit efficiency	0.625%	2.5%	3.125%	3.125%	7.134%

## Data Availability

The data that support the findings of this study are available from the corresponding author upon reasonable request.

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
