# Peer review of "A Semi-Quantum Private Comparison Base on W-States"

_entropy, 2023, doi:10.3390/e25091269_

Round 1

Reviewer 1 Report

With the GHZ three-particle entangled state, the authors offered a more efficient semi-quantum privacy comparison method and the scheme can resist third-party, measurement, and entanglement attacks.

The introduction should be more logically and more recent works should be considered. Advanced Quantum Technologies, 10.1002/qute.202300097. Quantum Information Processing,  Quantum Information Processing, 2021, 20(3): 124. Physica Scripta, 10.1088/1402-4896/acb61f.

Please explain the results theoretically.

It would be better to simulate the protocol in quantum platform. 

The expression can be polished.

Author Response

With the GHZ three-particle entangled state, the authors offered a more efficient semi-quantum privacy comparison method and the scheme can resist third-party, measurement, and entanglement attacks.

The introduction should be more logically and more recent works should be considered. Advanced Quantum Technologies, 10.1002/qute.202300097. Quantum Information Processing,  Quantum Information Processing, 2021, 20(3): 124. Physica Scripta, 10.1088/1402-4896/acb61f.

Thanks for this valuable comment and we have revised our paper carefully. In the introduction, we have made logical adjustments to the language, supplemented some contents, and added the latest research results in the introduction of the research status. For your convenience, we have made comments in the paper.

Please explain the results theoretically.

Thanks for your earnest comment and we have changed and adjusted this section, The result of this work and its safety analysis are supplemented and annotated in the paper.

It would be better to simulate the protocol in quantum platform. 

Thanks for your kind comment. However, the innovation of this work lies in the clever design of the protocol, such as key negotiation for different operation situations, and relevant circuit design is not carried out, so we did not carry out the experimental part of the circuit simulation.

Reviewer 2 Report

I have reviewed the article named A Semi-Quantum Private Comparison Base on GHZ-states and  W-states.  In the paper, it appears that the authors have proposed a more efficient semi-quantum privacy comparison method based on GHZ-states and W-states.

1.      In the section introducing semi-quantum privacy comparison and the basics of quantum information, it may be helpful to include more diagrams or examples to aid readers in understanding the concepts.

 2. In section 3, the explanation of the protocol's steps and contents could be more detailed and clear to help readers better understand and implement the protocol.

3. In section 4, I think that the security analysis proofs could be more comprehensive and detailed, more formal language and symbols could be used to increase the rigor of the proofs.

4. I suggest the author to introduce recently process in fundamental quantum device. Such as the device in the review Frontiers of Physics 17 (4), 42201

I think the language should be modified

Author Response

  1. In the section introducing semi-quantum privacy comparison and the basics of quantum information, it may be helpful to include more diagrams or examples to aid readers in understanding the concepts.

Thanks for this valuable comment and we have revised our paper carefully. We provide the necessary supplementary explanation of the basics through formulas and figures, and make comments.

  1. In section 3, the explanation of the protocol's steps and contents could be more detailed and clear to help readers better understand and implement the protocol.

Thanks for your earnest comment and we have made necessary supplementary explanations to this part to eliminate its possible ambiguity and so on.

  1. In section 4, I think that the security analysis proofs could be more comprehensive and detailed, more formal language and symbols could be used to increase the rigor of the proofs.

Thanks for this kind comment and We have carried out supplementary analysis in the security analysis section to make the security analysis more complete, and made comments in the paper for your convenience.

  1. I suggest the author to introduce recently process in fundamental quantum device. Such as the device in the review Frontiers of Physics 17 (4), 42201

Thank you for your kind reminder, We have updated this work in the introduction.

Reviewer 3 Report

This manuscript presents a semi-quantum privacy comparison method based on GHZ three-particle entangle states and has provided security and efficiency analyses. The work presented is of interest to the quantum community, but the language is incomprehensible. Here are some of the problems leading up to the point where the main technical work reaches a point that the presented method is not executable.

Line 31: "... constantly broken".

Sentence along seems fine but statement does not reflect facts. Maybe the authors are attempting to say "repeatedly broken".

Line 32: "Quantum privacy comparison ... applied as resistance to quantum attacks".

Grammar error. Not even a sentence.

Line 33: "As an indivisible smallest particle, quantum ...".

Basic fact error. Quantum is not a particle.

Line 73: "the participants 3-particles"

Basic language error.

Line 73: "The W-states and GHZ-states were used to distribute to the participants 3-particles, they are denoted by ..."

Need "and" before "they are denoted by"

Line 116: "TP then divides these three entangled particles into three sequences"

What three particles? It contradicts with Line 123 "the number of particles to perform MEASURE operation is n for Alice or Bob"

Three sequences of what? From Line 118, seems three sequences of particle representations. Then what is a representation?

Line 119: "sequenceto"

 Typo or what?

Line 122: "Step 3: Alice and Bob randomly perform both MEASURE and REFLECT ..."

This is ambiguous.  Does it mean "It has a 50% probability (or more generally a probably p following some probability distribution) that Alice and Bob perform both MEASURE and REFLECT and a 50% probability (or probability of 1-p) that they do nothing"?

The presentation at this line renders a technical ambiguity that makes the method inexecutable, and the reviewer is forced to recommend a complete rewrite of the manuscript.

The language needs improvement.

Author Response

Thanks for your earnest comment, As for the details of the description, we have checked the whole text and supplemented or modified it to make it clear and easier to understand; As for the basic errors, we have modified and checked and modified similar problems in the full text; For grammar or editing errors, we have made corrections; In order to facilitate re-examination, the amendments have been annotated in the annexed manuscript. Thank you again for your patience and care.

Reviewer 4 Report

The proposal covers the privacy problem of quantum multiparty communication usable in secure computation. The research subject is well explained and justified.

The authors justify the relevance of solving the proposed problem in the research field.  Abstract presents the problem and the approach, some examples of applications and the research results.

“Introduction” chapter contains a good state of the art and the related works linked to appropriate references.

The proposal is well linked and integrated in the literature of domain.

The methodology (i.e the theoretical background) is clear and well explained.

The paper is well structured, and the approach is formally described.

The communication algorithm is sustained by the proposed formulas.

The analysis of the achieved security and the algorithm efficiency are well performed. A comparison of the current method and similar methods included in references is given.   

The conclusions are based on the provided results. 

I have some recommendations: please specify the acronym GHZ and verify the lines 121 and 255.

Author Response

Thanks for this kind comment and we will check the grammar of the articles and pay more attention to English writing in the subsequent work.
